# Decreased Antibiotic Consumption Coincided with Reduction in Bacteremia Caused by Bacterial Species with Respiratory Transmission Potential during the COVID-19 Pandemic

**DOI:** 10.3390/antibiotics11060746

**Published:** 2022-05-31

**Authors:** Vincent Chi-Chung Cheng, Shuk-Ching Wong, Simon Yung-Chun So, Jonathan Hon-Kwan Chen, Pui-Hing Chau, Albert Ka-Wing Au, Kelvin Hei-Yeung Chiu, Xin Li, Patrick Ip, Vivien Wai-Man Chuang, David Christopher Lung, Cindy Wing-Sze Tse, Rodney Allan Lee, Kitty Sau-Chun Fung, Wing-Kin To, Raymond Wai-Man Lai, Tak-Lun Que, Janice Yee-Chi Lo, Kwok-Yung Yuen

**Affiliations:** 1Infection Control Team, Queen Mary Hospital, Hong Kong West Cluster, Hong Kong, China; vcccheng@hku.hk (V.C.-C.C.); shchwong@hku.hk (S.-C.W.); 2Department of Microbiology, Queen Mary Hospital, Hong Kong, China; syc687@ha.org.hk (S.Y.-C.S.); jonchk@hku.hk (J.H.-K.C.); chy731@ha.org.hk (K.H.-Y.C.); 3School of Nursing, Li Ka Shing Faculty of Medicine, The University of Hong Kong, Hong Kong, China; phchau@graduate.hku.hk; 4Centre for Health Protection, Department of Health, Hong Kong, China; albert_au@dh.gov.hk (A.K.-W.A.); janicelo@dh.gov.hk (J.Y.-C.L.); 5Department of Microbiology, Li Ka Shing Faculty of Medicine, The University of Hong Kong, Hong Kong, China; xinli@hku.hk; 6Department of Paediatrics and Adolescent Medicine, Li Ka Shing Faculty of Medicine, The University of Hong Kong, Hong Kong, China; patricip@hku.hk; 7Department of Paediatrics and Adolescent Medicine, The Hong Kong Children’s Hospital, Hong Kong, China; 8Quality & Safety Division, Hospital Authority, Hong Kong, China; chuangwm@ha.org.hk (V.W.-M.C.); wmlai@ha.org.hk (R.W.-M.L.); 9Department of Pathology, Queen Elizabeth Hospital, Hong Kong, China; lungdc@ha.org.hk; 10Department of Pathology, The Hong Kong Children’s Hospital, Hong Kong, China; 11Department of Pathology, Kwong Wah Hospital, Hong Kong, China; tsewsc@ha.org.hk; 12Department of Pathology, Pamela Youde Nethersole Eastern Hospital, Hong Kong, China; leear@ha.org.hk; 13Department of Pathology, United Christian Hospital, Hong Kong, China; fungsck@ha.org.hk; 14Department of Pathology, Princess Margaret Hospital, Hong Kong, China; towk@ha.org.hk; 15Department of Pathology, Prince of Wales Hospital, Hong Kong, China; 16Department of Pathology, Tuen Mun Hospital, Hong Kong, China; quetl@ha.org.hk

**Keywords:** antibiotic consumption, bacteremia, transmission, COVID-19

## Abstract

Nonpharmaceutical interventions implemented during the COVID-19 pandemic (2020–2021) have provided a unique opportunity to understand their impact on the wholesale supply of antibiotics and incidences of infections represented by bacteremia due to common bacterial species in Hong Kong. The wholesale antibiotic supply data (surrogate indicator of antibiotic consumption) and notifications of scarlet fever, chickenpox, and tuberculosis collected by the Centre for Health Protection, and the data of blood cultures of patients admitted to public hospitals in Hong Kong collected by the Hospital Authority for the last 10 years, were tabulated and analyzed. A reduction in the wholesale supply of antibiotics was observed. This decrease coincided with a significant reduction in the incidence of community-onset bacteremia due to *Streptococcus pyogenes*, *Streptococcus pneumoniae*, *Haemophilus influenzae*, and *Neisseria meningitidis*, which are encapsulated bacteria with respiratory transmission potential. This reduction was sustained during two pandemic years (period 2: 2020–2021), compared with eight pre-pandemic years (period 1: 2012–2019). Although the mean number of patient admissions per year (1,704,079 vs. 1,702,484, *p* = 0.985) and blood culture requests per 1000 patient admissions (149.0 vs. 158.3, *p* = 0.132) were not significantly different between periods 1 and 2, a significant reduction in community-onset bacteremia due to encapsulated bacteria was observed in terms of the mean number of episodes per year (257 vs. 58, *p* < 0.001), episodes per 100,000 admissions (15.1 vs. 3.4, *p* < 0.001), and per 10,000 blood culture requests (10.1 vs. 2.1, *p* < 0.001), out of 17,037,598 episodes of patient admissions with 2,570,164 blood culture requests. Consistent with the findings of bacteremia, a reduction in case notification of scarlet fever and airborne infections, including tuberculosis and chickenpox, was also observed; however, there was no reduction in the incidence of hospital-onset bacteremia due to *Staphylococcus aureus* or *Escherichia coli*. Sustained implementation of non-pharmaceutical interventions against respiratory microbes may reduce the overall consumption of antibiotics, which may have a consequential impact on antimicrobial resistance. Rebound of conventional respiratory microbial infections is likely with the relaxation of these interventions.

## 1. Introduction

The outbreak of severe acute respiratory syndrome (SARS) due to the SARS coronavirus 1 (SARS-CoV-1) in 2003, and pandemic influenza A H1N1 in 2009, highlighted the importance of infection control and preparedness in public health responses in hospital and community settings [1,2]. With the implementation of proactive infection control measures, including the promotion of directly observed hand hygiene for hospitalized patients before meal and medication rounds [3,4], and patient empowerment in hand hygiene [5], we have successfully prevented and controlled hospital outbreaks caused by epidemiologically important respiratory and gastrointestinal viruses in the Hong Kong Special Administrative Region of China (HKSAR) [6]; therefore, when the coronavirus disease 2019 (COVID-19) pandemic emerged, hospital infection control measures were enforced in accordance with the consensus recommendation [7] to minimize nosocomial SARS-CoV-2 transmission [8,9]. Moreover, early implementation of universal masking, in addition to enhanced hand hygiene practice in our hospitals, further achieved zero nosocomial transmission of other respiratory viruses [10,11]. Similarly, community implementation of nonpharmaceutical interventions has also significantly reduced the incidence of respiratory viruses in Hong Kong [12], South Korea [13], Japan [14], Singapore [15], Germany [16], Canada [17], the United States [18], and Brazil [19]. These findings were expected with the implementation of universal masking, hand hygiene, social distancing and intermittent school closures in the community during the COVID-19 pandemic. With these heightened infection control measures in hospital and community settings during the past two years, we would expect a reduction of infections due to community-acquired and hospital-acquired bacteria that are transmitted through contact or droplet routes, which may further lead to a reduction of antibiotic consumption in community and healthcare settings [20,21]. Though reduction of invasive disease due to *Streptococcus pneumoniae*, *Haemophilus influenzae*, or *Neisseria meningitidis* has been reported during the first year of the COVID-19 pandemic [22,23,24,25], the sustainability of such a reduction is still uncertain.

The two years of the COVID-19 pandemic provided us with a unique opportunity to investigate the change in the wholesale supply of antibiotics and the incidence of community or hospital acquired bacteremia. When we compared the findings in these two pandemic years with the preceding years, we observed a reduction in the wholesale supply of antibiotics in the community setting, which coincided with a significant reduction in bacteremia, due to encapsulated bacteria, with the potential of respiratory transmission, including *Streptococcus pyogenes*, *S. pneumoniae*, *H. influenzae*, and *N. meningitidis*, but not methicillin-sensitive *Staphylococcus aureus* (MSSA), methicillin-resistant *S. aureus* (MRSA), and *Escherichia coli*. This observation may have implications on recommendations of infection control and public health measures.

## 2. Methods

### 2.1. Setting

This is a retrospective study on the change in wholesale antibiotic supply and occurrence of community-onset bacteremia due to common bacteria with or without the potential for droplet or aerosol transmission among hospitalized patients in all 43 public hospitals managed by the Hospital Authority in the HKSAR before and during COVID-19. These hospitals provide 90% of inpatient services for our 7.5 million population. The baseline period before the outbreak of COVID-19 was defined as period 1 (2012 to 2019), whereas the two-year period of the COVID-19 pandemic was defined as period 2 (2020 to 2021).

### 2.2. Wholesale Supply of Antibiotics in Hong Kong

The data was obtained from the Drug Office of the Department of Health, collected from all licensed drug wholesalers in Hong Kong. The data on antimicrobial supply to different sectors including private doctors, community pharmacists, private hospitals, and public hospitals are used as a surrogate indicator of antibiotic consumption, which is expressed as defined daily doses per 1000 inhabitants per day [26].

### 2.3. Data Source

The episode-based records for all inpatients, including demographic information and number of blood culture requests, are retrieved from the Clinical Data Analysis and Reporting System (CDARS), an electronic database of health records managed by the Hospital Authority. A unique hospital number (HN) is given for each hospital admission. Episodes of hospitalization with a positive blood culture of selected pathogens for 10 years (from 1 January 2012 to 31 December 2021) were retrieved from CDARS. The selected microorganisms included pathogens with potential for respiratory transmission (*S. pyogenes*, *S. pneumoniae*, *H. influenzae*, and *N. meningitidis*), and pathogens commonly identified in blood cultures including MSSA, MRSA, and *E. coli*, which are not considered to have potential for respiratory transmission, and they indirectly served as controls. An episode of community-onset and hospital-onset bacteremia was defined as positive blood culture isolation of the selected pathogens at ≤2 days and >2 days of hospitalization, respectively. If a patient had more than one episode of community-onset or hospital-onset bacteremia due to the same pathogen with the same HN, only the first episode was counted.

### 2.4. Overview of COVID-19 in Hong Kong

When the outbreak of community-acquired pneumonia of unknown etiology in mainland China (subsequently identified to be SARS-CoV-2) was initially announced on 31 December 2019 [27], and a progressive escalation of infection control measures was implemented to minimize the risk of nosocomial transmission of SARS-CoV-2 [28,29,30]. Quarantine measures were administered to inbound travelers to minimize the risk of importation of SARS-CoV-2 [31,32]. Extensive contact tracing was performed to control the spread of SARS-CoV-2 in the community [33]. Universal masking commenced in community and healthcare settings [11,34]. Diagnosis of COVID-19 was confirmed by a reverse transcription polymerase chain reaction using combined nasal and throat swabs collected by trained staff in the hospitals, community treatment facilities, and community testing centers [35,36]. Deep throat saliva was subsequently accepted as an alternative specimen for diagnosis [37].

### 2.5. Number of Patient Admissions and Blood Culture Requests before and during COVID-19

The number of patient admissions and blood culture requests from 2012 to 2021 among all public hospitals in Hong Kong were retrieved. The mean number of patient admissions and blood culture requests per year, and the rate of blood culture requests per 1000 patient admissions, were analyzed.

### 2.6. Community-Onset Bacteremia before and during COVID-19

The mean number of patient episodes of community-onset bacteremia associated with the selected pathogens and their rates per 100,000 patient admissions and per 10,000 blood culture requests were compared between period 1 and period 2. The change of mean episodes of community-onset bacteremia of the selected pathogens among patients ≤12 or >12 years of age was also analyzed. The finding concerning the incidence of *S. pyogenes* bacteremia was independently validated using the statutory notifications on scarlet fever, which is caused by *S. pyogenes*. Notifications of infections due to airborne pathogens, including tuberculosis and chickenpox, during the same period were also analyzed as surrogate positive controls [38].

### 2.7. Hospital-Onset Bacteremia before and during COVID-19

The mean number of patient episodes of hospital-onset bacteremia among the selected pathogens and the rates per 100,000 patient admissions and per 10,000 blood culture requests were compared between period 1 and 2.

### 2.8. Statistical Analysis

Differences in the magnitude of the wholesale supply of antibiotics, expressed in terms of being defined by daily doses per 1000 inhabitants per day, as well as the community-onset and hospital-onset bacteremia due to *S. pyogenes*, *S. pneumoniae*, *H. influenzae*, MSSA, MRSA, and *E. coli* in terms of number and rates per 100,000 patient admissions, and per 100,000 blood culture requests, were evaluated between period 1 (2012 to 2019) and period 2 (2020 to 2021) using Poisson Regression. All statistical analyses were performed using IBM SPSS Statistics (version 26). A two-sided *p*-value of <0.05 was considered statistically significant.

## 3. Results

### 3.1. Overview of COVID-19 in Hong Kong

The first case of COVID-19 in Hong Kong was an incoming traveler reported on 23 January 2020. Until 31 December 2021, there was a total of 12,655 laboratory-confirmed COVID-19 patients, with a median of 257 cases per month (range: 13 to 2532 cases). All the laboratory confirmed COVID-19 patients were admitted to the airborne infection isolation facilities in public hospitals, or the community isolation or treatment facilities. There were 6113 (48.3%) males. The median age was 43 years (range: 12 days to 100 years). The epidemic curve of COVID-19 cases is shown in Figure 1.

### 3.2. Wholesale Supply of Antibiotics in Hong Kong

Despite the progressive reduction of antibiotic supply to community pharmacies, the supply of antibiotics to private hospitals remained at a similar level from 2017 to 2019, whereas the consumption of antibiotics by private doctors and the Hospital Authority peaked in 2019 (Figure 2). During the COVID-19 pandemic, a reduction of antibiotic supplies to private doctors (39.0%), community pharmacies (47.9%), private hospitals (35.1%), and the Hospital Authority (12.5%) was observed between 2019 and 2021 in Hong Kong. The mean amount of the wholesale supply of antibiotics, expressed in terms of being defined by daily doses per 1000 inhabitants per day, that was supplied to private doctors, was significantly lower during the COVID-19 pandemic (2020–2021) compared with the preceding years (2014 to 2019) (10.40 vs. 6.74, *p* < 0.001). Similarly, the corresponding figure for community pharmacies was also significantly lower (2.75 vs. 0.91, *p* < 0.001).

### 3.3. Number of Patient Admissions and Blood Culture Requests before and during COVID-19

From 2012 to 2021, there were 17,037,598 episodes of patient admissions to public hospitals. A comparison between the mean number of patient admissions per year before COVID-19 (period 1, 2012 to 2019) and during COVID-19 (period 2, 2020 to 2021) showed no significant difference (1,704,079 vs. 1,702,484, *p* = 0.985). During the study period, there was a total of 2,570,164 blood culture requests from hospitalized patients. There was no significant difference in the mean number of blood culture requests per year (253,896 vs. 269,500, *p* = 0.225) and the mean number of blood culture requests per 1000 patient admissions (149.0 vs. 158.3, *p* = 0.132) between period 1 and period 2 (Figure 3). Of the 2,570,164 blood culture requests from hospitalized patients, 93,497 (3.6%) patients had bacteremia due to the seven selected pathogens (*S. pyogenes*, *S. pneumoniae*, *H. influenzae*, *N. meningitidis*, MSSA, MRSA, and *E. coli*). Among the 93,497 patients, 71,923 (76.9%) had community-onset bacteremia, whereas 21,574 (23.1%) had hospital-onset bacteremia.

### 3.4. Community-Onset Bacteremia before and during COVID-19

From 2012 to 2021, among the 71,923 episodes of community-onset bacteremia due to selected pathogens, 56,937 (79.2%) and 14,986 (20.8%) episodes were reported in period 1 and period 2, respectively (Figure 4). There was no significant difference between the mean number of episodes of community-onset bacteremia per year in period 1 and period 2 (7117 vs. 7493, *p* = 0.186). Of these 71,923 episodes, 2172 (3.0%) episodes were due to encapsulated bacteria with potential for respiratory transmission (*S. pyogenes*: 638 episodes; *S. pneumoniae*: 1278 episodes, *H. influenzae*: 222 episodes, *N. meningitidis*: 34 episodes), whereas the remaining 69,715 (97.0%) episodes were due to MSSA (6959), MRSA (4851), and *E. coli* (57,941). For bacteremia due to encapsulated bacteria, the annual episodes of community-onset bacteremia decreased from the maximum of 302 (2014) to 68 in 2020 and 47 in 2021 (Figure 5). The incidence of community-onset bacteremia due to the encapsulated bacteria per 100,000 patient admissions (15.1 vs. 3.4, *p* < 0.001) and per 10,000 blood culture requests (10.1 vs. 2.1, *p* < 0.001) also decreased significantly between periods 1 and 2. Although the mean episodes of community-onset bacteremia due to the encapsulated bacteria per year significantly decreased from period 1 to period 2 (257 vs. 58, *p* < 0.001), the mean episodes of community-onset bacteremia due to MSSA and MRSA per year in period 2 were also significantly higher than period 1 (1154 vs. 1288, *p* = 0.001).The changes and statistical analysis in the mean episodes of community-onset bacteremia of the individual pathogens per 100,000 patient admissions and per 10,000 blood culture requests, as well as the notifications of scarlet fever, tuberculosis, and chickenpox as surrogate controls, are illustrated in Table 1. The changes in mean episodes of community-onset bacteremia of the individual pathogens among patients ≤12 or >12 years of age are shown in Table 2. For those ≤12 years of age, a significant decrease in mean episodes of bacteremia due to *S. pyogenes* and *S. pneumoniae* was observed, whereas for those >12 years of age, a significant decrease in mean episodes of bacteremia due to *S. pyogenes*, *S. pneumoniae* and *H. influenzae* was observed in period 2.

### 3.5. Hospital-Onset Bacteremia before and during COVID-19

From 2012 to 2021, a total of 18,418 episodes of hospital-onset bacteremia of the selected pathogens were retrieved during our study period, of which 14,593 (79.2%) and 3825 (20.8%) episodes were reported in period 1 and period 2, respectively. No significant difference between the mean episode of hospital-onset bacteremia per year among the seven selected pathogens was observed between period 1 and period 2 (1824 vs. 1913, *p* = 0.305). For the pathogens with potential for respiratory transmission (*S. pyogenes*, *S. pneumoniae*, *H. influenzae*), calculated as a whole, the mean episode of hospital-onset bacteremia (12 vs. 3, *p* < 0.001), per 100,000 patient admissions (0.72 vs. 0.15, *p* < 0.001), and per 10,000 blood culture requests (0.48 vs. 0.09, *p* < 0.001), decreased significantly from period 1 to period 2 (Table 3). There was no hospital-onset *N. meningitidis* bacteremia during both period 1 and period 2.

## 4. Discussion

During the first two years of the COVID-19 pandemic, the wholesale supply of antibiotics as a surrogate indicator of antibiotic consumption has markedly decreased, and this coincided with a significant decrease in the incidence of community-onset bacteremia due to *S. pyogenes*, *S. pneumoniae*, *H. influenzae*, and *N. meningitidis* in terms of the number of isolates per 100,000 patient admissions and per 10,000 blood culture requests; however, this phenomenon was not observed for MSSA, MRSA, and *E. coli* bacteremia compared with the 8-year pre-pandemic period, according to our 10-year territory-wide surveillance of over 2 million blood cultures requested amongst over 17 million episodes of patient admissions in all public hospitals in Hong Kong. *S. aureus* and *E. coli* are well known to be transmitted through direct and indirect contact of patients, food, and environment, including contaminated banknotes [39], whereas *S. pyogenes*, *S. pneumoniae*, *H. influenzae*, and *N. meningitidis* are long believed to be transmitted by droplets or direct/indirect contact with respiratory secretion [40,41]. For *S. pneumoniae*, studies have observed that progression from benign carriage to invasive infection often occurs only a few days after the acquisition of an upper airway infection [42,43,44]. Furthermore, airborne transmission of *S. pyogenes* has been postulated as being an outbreak of scarlet fever in school children [45], whereas airborne transmission of *S. pneumoniae* was only demonstrated among closely housed ferrets that were co-infected with the influenza A virus [46]. Some evidence of droplet transmission of *S. pneumoniae* was reported in outbreak settings [47,48]. Airborne contamination by *H. influenzae* has been demonstrated in children’s daycare centers [49]. Inhalation of large airborne droplets produced by coughing or sneezing from colonized individuals was hypothesized to be the transmission route of *N. meningitidis* [50]. Overall, the epidemiological and experimental evidence of respiratory transmission of these encapsulated bacteria colonizing the upper airway is rather anecdotal and fragmented.

The marked increase in compliance with non-pharmaceutical interventions such as universal masking, hand hygiene, and social distancing measures, including intermittent school closures during the past two years of the COVID-19 pandemic, provided a unique opportunity to determine whether the incidence of these common pathogenic bacteria has sustainably changed, as first reported during the initial months of COVID-19 [22,23,24,25]. Despite these hygienic and social distancing measures, the incidence of community-onset or hospital-onset MSSA or *E. coli* bacteremia did not decrease during these two pandemic years. Notably, although the number of blood culture requests has not significantly changed, a significant decrease in the incidence of bacteremia due to *S. pyogenes*, *S. pneumoniae*, *H. influenzae*, and *N. meningitidis* occurred, which is believed to have potential for transmission through droplet or airborne routes. These findings suggested that universal masking and other non-pharmaceutical interventions may not only be useful for interrupting the transmission of SARS-CoV-2 or other respiratory viruses [12,13,14,15,16,17,18,19], but also for these encapsulated bacteria colonizing the upper respiratory tract [22,23,24,25].

Since the nasal carriage of *S. aureus* and the gut carriage rate of *E. coli* are around 30% and 100% in the adult population, respectively [51], *S. aureus* and *E. coli* bacteremia may have originated from endogenous flora, rather than having been recently acquired during the pandemic [52]. This may partially explain the finding that the incidence of both *S. aureus* and *E. coli* bacteremia were not significantly decreased despite the likely improved compliance with hand hygiene in the community during these two pandemic years. Furthermore, full compliance with universal masking by healthcare workers is much easier than compliance with hand hygiene. It is not inconceivable that the incidence of hospital-onset *S. aureus* and *E. coli* bacteremia has not changed significantly during this period when the hospital workload has markedly increased during the pandemic years.

Frequent school closures during the COVID-19 pandemic may also explain the significantly lower incidence of bacteremia due to encapsulated bacteria in adults, in an indirect manner. Experience during seasonal influenza epidemics in many communities suggested that influenza usually starts amongst school children with higher viral loads and shedding, who, in turn, infect their family members, including the elderly. Encapsulated bacteria may also spread from school children to family members through similar pathway. With the frequent school closures, it may lead to a lower bacterial carriage rate in school children, and hence, a lower rate of transmission to family members. Furthermore, the incidence of influenza and other respiratory viruses have dramatically decreased in different populations over the past two years [12,13,14,15,16,17,18,19]. The influenza virus alters the lung epithelial cells which become highly susceptible to adherence, invasion, and induction of disease by *S. pneumoniae*. It is possible that influenza or other respiratory viruses may expose or upregulate receptors for virus entry, induce a lethal synergism of severe inflammation and cytotoxic damage between bacteria and viruses, or diminish the ability of the host to clear encapsulated bacteria [53,54,55]. Despite clinical and experimental evidence that the influenza A virus is predisposed to *S. aureus* coinfection or superinfection, the virtually complete absence of seasonal influenza in these two pandemic years were not associated with a lower rate of community- or hospital-onset *S. aureus* bacteremia. One of the explanations of this observation could be due to previous findings of a low proportion of bacteremia contributed by *S. aureus* superinfecting influenza patients [56].

The decrease in community-onset bacteremia, especially due to *S. pneumoniae* and *H. influenzae*, was unlikely to be related to our childhood vaccination program as the pneumococcal 7-valent conjugate vaccine and pneumococcal 13-valent conjugate vaccine were introduced in 2009 and 2011, respectively, whereas the *H. influenzae* B vaccine is available in our private medical service, but has not yet been included in our government-funded childhood vaccination program; therefore, the significant reduction in community-onset bacteremia of encapsulated bacteria during the COVID-19 pandemic is more likely related to infection control measures instead of the implementation of vaccination. It is interesting to note that there was a significant increase in community-onset MSSA and MRSA bacteremia during the pandemic. This could be attributed to the delayed seeking of medical management as the general public were apprehensive of catching COVID-19 during visits to healthcare facilities.

Although the exact contribution of universal masking, enhanced hand hygiene, social distancing, and school closures, to the decrease in bacteremia due to these four encapsulated bacteria with respiratory transmission potential, cannot be ascertained, the concomitant reduction of incidences of classical airborne infections, including tuberculosis and chickenpox during the COVID-19 pandemic, suggests that universal masking might play a significant role. Our findings may also predict the resurgence of all these bacteremic, and perhaps, nonbacteremic diseases, due to these encapsulated bacteria when the nonpharmaceutical interventions for the control of the COVID-19 pandemic were lifted after the Omicron wave. It is likely that in the short-term, less stable colonization by these encapsulated bacteria in the upper airway will be more easily affected by these COVID-19 control measures than in the long-term. Moreover, more stable colonization by *S. aureus* and *E. coli* will occur, as demonstrated by the higher prevalence of the bacteremia of *S. aureus* and *E. coli* in this study, when compared with these four encapsulated bacteria with the potential for respiratory transmission.

The significant reduction of community-onset bacteremia due to encapsulated bacteria coincided with the drop in overall antibiotic sales in Hong Kong, especially at the community healthcare level. The reduction in this surrogate indicator of antibiotic consumption during the COVID-19 pandemic is consistent with the territory-wide study conducted in mainland China [21]; however, when the antibiotic consumption in COVID-19 patients was analyzed, a large proportion of patients were treated with antibiotics even when these patients were having mild and moderate COVID-19 symptoms, with no clinical evidence of bacterial co-infection [57]. Moreover, the number of prescriptions was significantly higher than the estimated prevalence of bacterial co-infection requiring antibiotic treatment in a meta-analysis [58]. Over prescription of antibiotics may lead to the emergence of multi-drug resistant organisms and hospital outbreaks [59,60]. Inappropriate antibiotic prescriptions given to COVID-19 patients may be augmented by the perceived burden and severity of SARS-CoV-2 infection [61], as well as the lack of enforcement of the antibiotic stewardship program due to the overloading of the healthcare system during the pandemic [62]. Our reduction of antibiotic consumption may also be attributed to the relatively low number of confirmed COVID-19 patients before the Omicron wave, and the decrease in community-acquired infection due to respiratory viruses and encapsulated bacteria as a result of the enhancement of infection control and non-pharmaceutical interventions.

The findings of our study highlight that wearing masks in the community is likely to be an important measure that contributes to the prevention of life-threatening infections caused by encapsulated bacteria with potential for respiratory transmission. Our study is the first to report a significant decrease in *S. pyogenes* bacteremia during these two pandemic years. Since there is no *S. pyogenes* vaccine, and scarlet fever has become highly prevalent since 2011 [63,64], infection control measures remain key for the prevention and control of invasive *S. pyogenes* infections.

There are several limitations of this study. Firstly, we do not completely understand the finding of a significant increase in community-onset MRSA bacteremia during the COVID-19 pandemic, despite the improvement of infection control awareness among the public; however, we also observed an increasing trend of MRSA colonization among residents in the residential care homes for the elderly (RCHE) during the COVID-19 pandemic [65,66]. The increase in the burden of MRSA in RCHE, and in community-onset MRSA bacteremia in hospitals, deserves further investigations. Secondly, we did not include data from January 2022 when severe community outbreaks due to the co-circulation of the Omicron subvariant BA.2 and Delta variant of SARS-CoV-2 occurred in Hong Kong [67,68]; however, we believe that the more extensive adoption of infection control measures against the Omicron variant would lead to an even greater decline in the incidence of community-onset bacteremia due to these encapsulated bacteria.

## 5. Conclusions

In summary, the implementation of nonpharmacological interventions during the last two COVID-19 pandemic years in Hong Kong has markedly reduced the incidence of acute respiratory viruses and bacterial infections transmitted by droplet and airborne routes. This change is most evident in the incidence of community-onset bacteremia due to *S. pyogenes*, *S. pneumoniae*, *H. influenzae*, and *N. meningitidis*, and is independently confirmed by the marked decrease in the notification of scarlet fever, chickenpox, and tuberculosis, which are largely droplet or airborne infections. Notably, this change has coincided with the decrease in our wholesale supply of antibiotics. Our findings suggested that nonpharmaceutical measures in reducing the incidence of respiratory infections may be one of the strategies to control the amount of antibiotic consumption in humans, which may impact the burden of antibacterial resistance in the long run. However, a major rebound in the incidences of these respiratory pathogens and antibiotic consumption is highly likely when these measures are relaxed.

## Figures and Tables

**Figure 1 antibiotics-11-00746-f001:**
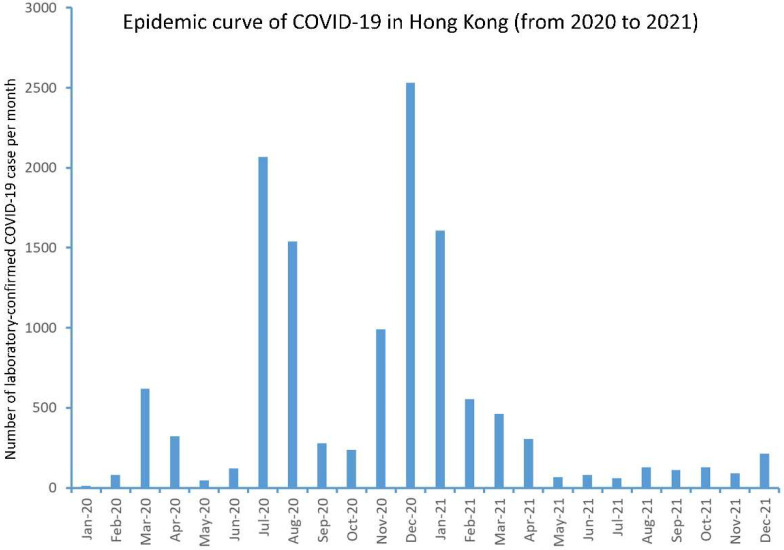
Epidemic curve of laboratory-confirmed COVID-19 cases in Hong Kong (from 2020 to 2021).

**Figure 2 antibiotics-11-00746-f002:**
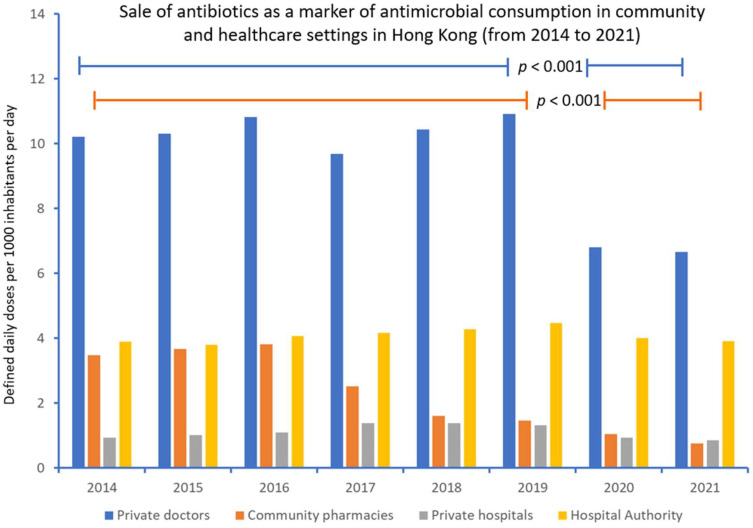
Supply of antibiotics as a marker of antimicrobial consumption in community and healthcare settings in Hong Kong (from 2014 to 2021).

**Figure 3 antibiotics-11-00746-f003:**
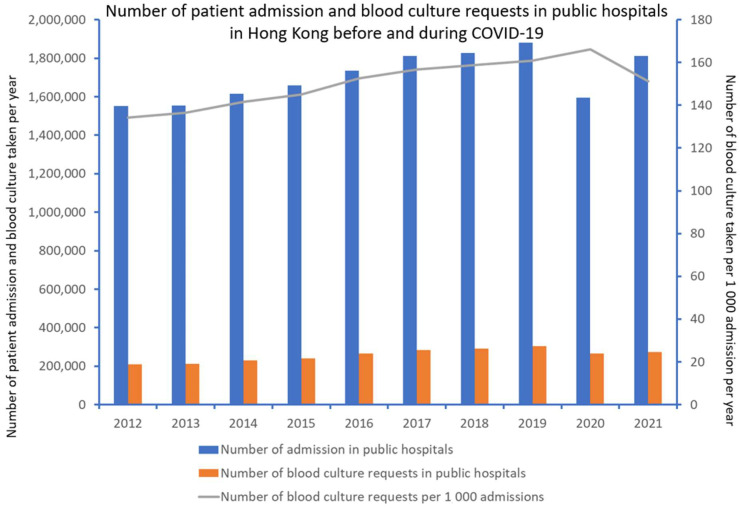
Number of patient admission and blood culture requests in public hospitals in Hong Kong before and during COVID-19.

**Figure 4 antibiotics-11-00746-f004:**
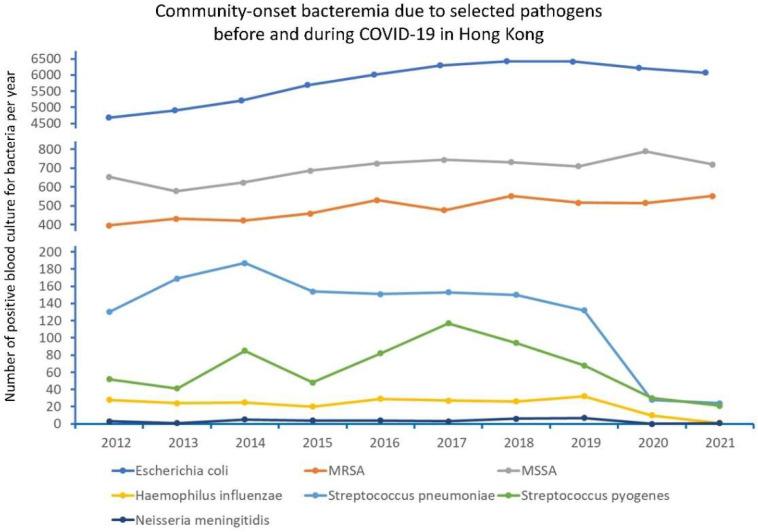
Community-onset bacteremia due to selected pathogens before and during COVID-19 in Hong Kong. Note: the selected microorganisms included pathogens with potential respiratory transmission (*Streptococcus pyogenes, Streptococcus pneumoniae, Haemophilus influenzae,* and *Neisseria meningitidis*), and pathogens commonly identified in the blood culture including methicillin-sensitive *Staphylococcus aureus* (MSSA), methicillin-resistant *S. aureus* (MRSA), and *Escherichia coli*, were used as controls. Statistical analysis is shown in Table 1.

**Figure 5 antibiotics-11-00746-f005:**
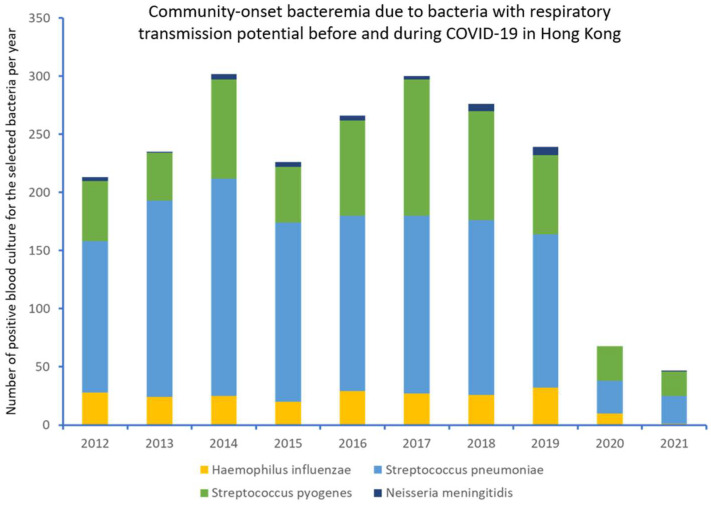
Community-onset bacteremia due to pathogens with respiratory transmission potential before and during COVID-19 in Hong Kong. Note: the pathogens with potential respiratory transmission include *Streptococcus pyogenes*, *Streptococcus pneumoniae*, *Haemophilus influenzae* and *Neisseria meningitidis*.

**Table 1 antibiotics-11-00746-t001:** Community-onset bacteremia of selected pathogens and notifications of scarlet fever, tuberculosis, and chickenpox before and during the COVID-19 pandemic in Hong Kong ^a^.

	From 2012 to 2019 (Period 1)	From 2020 to 2021 (Period 2)	*p* Value
*Streptococcus pyogenes*			
Total number of blood cultures	587	51	
Mean (range) blood cultures per year	73 (41–117)	26 (21–30)	<0.001
Per 100,000 patient admissions	4.31	1.50	<0.001
Per 10,000 blood culture requests	2.89	0.95	<0.001
*Streptococcus pneumoniae*			
Total number of blood cultures	1226	52	
Mean (range) blood cultures per year	153 (130–187)	26 (28–24)	<0.001
Per 100,000 patient admissions	8.99	1.53	<0.001
Per 10,000 blood culture requests	6.04	0.96	<0.001
*Haemophilus influenzae*			
Total number of blood cultures	211	11	
Mean (range) blood culture per year	26 (20–32)	6 (1–10)	0.007
Per 100,000 patient admissions	1.55	0.32	0.012
Per 10,000 blood culture requests	1.04	0.20	0.006
*Neisseria meningitidis*			
Total number of blood cultures	33	1	
Mean (range) blood cultures per year	4 (1–7)	1 (0–1)	0.004
Per 100,000 patient admissions	0.24	0.03	0.002
Per 10,000 blood culture requests	0.16	0.02	0.002
Methicillin-sensitive *Staphylococcus aureus*			
Total number of blood cultures	5451	1508	
Mean (range) blood cultures per year	681 (578–744)	754 (719–789)	0.020
Per 100,000 patient admissions	39.98	44.29	0.197
Per 10,000 blood culture requests	26.84	27.98	0.432
Methicillin-resistant *Staphylococcus aureus*			
Total number of blood cultures	3784	1067	
Mean (range) blood cultures per year	473 (396–552)	543 (515–552)	0.009
Per 100,000 patient admissions	27.76	31.34	<0.001
Per 10,000 blood culture requests	18.63	19.80	0.024
*Escherichia coli*			
Total number of blood cultures	45,645	12,296	
Mean (range) blood cultures per year	5706 (4681–6428)	6148 (6076–6220)	0.070
Per 100,000 patient admissions	334.80	361.10	0.176
Per 10,000 blood culture requests	224.70	228.10	0.516
Notifications of			
Scarlet fever			
Total number of notifications	12,567	351	
Mean (range) of notifications per year	1571	176	<0.001
Per 100,000 inhabitants	21.47	2.36	<0.001
Tuberculosis			
Total number of notifications	35,512	7397	
Mean (range) of notifications per year	4439 (4003–4858)	3699 (3656–3741)	<0.001
Per 100,000 inhabitants	60.66	49.66	<0.001
Chickenpox			
Total number of notifications	68,776	3577	
Mean (range) of notifications per year	8597 (6898–10,926)	1789 (1590–1987)	<0.001
Per 100,000 inhabitants	117.49	24.02	<0.001

^a^ Blood culture collected ≤ 2 days after hospital admission.

**Table 2 antibiotics-11-00746-t002:** Community-onset bacteremia of selected pathogens before and during the COVID-19 pandemic in Hong Kong according to age group ^a,b^.

	From 2012 to 2019 (Period 1)	From 2020 to 2021 (Period 2)	*p* Value
Patient aged ≤ 12 years			
*Streptococcus pyogenes*			
Total number of blood cultures	42	0	
Mean (range) blood cultures per year	5 (1–10)	0	NA ^c^
*Streptococcus pneumoniae*			
Total number of blood cultures	101	1	
Mean (range) blood cultures per year	13 (7–16)	0.5 (0–1)	<0.001
*Haemophilus influenzae*			
Total number of blood cultures	10	0	
Mean (range) blood cultures per year	1 (1–3)	0	NA ^c^
Methicillin-sensitive *Staphylococcus aureus*			
Total number of blood cultures	115	24	
Mean (range) blood cultures per year	14 (11–18)	12 (9–15)	0.337
Methicillin-resistant *Staphylococcus aureus*			
Total number of blood cultures	15	3	
Mean (range) blood cultures per year	2 (1–3)	2 (0–3)	0.755
*Escherichia coli*			
Total number of blood cultures	350	71	
Mean (range) blood cultures per year	44 (34–54)	36 (30–41)	0.079
Patient aged > 12 years			
*Streptococcus pyogenes*			
Total number of blood cultures	544	51	
Mean (range) number of blood cultures per year	68 (37–108)	26 (21–30)	<0.001
*Streptococcus pneumoniae*			
Total number of blood cultures	1125	51	
Mean (range) blood cultures per year	141 (119–175)	26 (23–28)	<0.001
*Haemophilus influenzae*			
Total number of blood cultures	201	11	
Mean (range) blood cultures per year	25 (20–31)	6 (1–10)	0.009
Methicillin-sensitive *Staphylococcus aureus*			
Total number of blood cultures	5333	1483	
Mean (range) blood cultures per year	667 (564–733)	742 (704–799)	0.021
Methicillin-resistant *Staphylococcus aureus*			
Total number of blood cultures	3768	1064	
Mean (range) blood cultures per year	471 (394–551)	532 (515–549)	0.007
*Escherichia coli*			
Total number of blood cultures	45,295	12,225	
Mean (range) blood cultures per year	5662 (4627–6390)	6113 (6046–6179)	0.065

^a^ Blood culture collected ≤ 2 days after hospital admission. ^b^ In light of the limited number of cases, *Neisseria meningitidis* was not analyzed by age. ^c^
*p*-value not available due to zero cases in period 2.

**Table 3 antibiotics-11-00746-t003:** Hospital-onset bacteremia of selected pathogens before and during the COVID-19 pandemic in Hong Kong ^a^.

	From 2012 to 2019 (Period 1)	From 2020 to 2021 (Period 2)	*p* Value
*Streptococcus pyogenes*			
Total number of blood cultures	19	1	
Mean (range) blood cultures per year	2 (0–6)	1 (0–1)	0.043
Per 100,000 patient admissions	0.14	0.03	0.053
Per 10,000 blood culture requests	0.09	0.02	0.036
*Streptococcus pneumoniae*			
Total number of blood cultures	50	3	
Mean (range) blood cultures per year	6 (3–11)	2 (1–2)	<0.001
Per 100,000 patient admissions	0.37	0.09	<0.001
Per 10,000 blood culture requests	0.25	0.06	<0.001
*Haemophilus influenzae*			
Total number of blood cultures	29	1	
Mean (range) blood cultures per year	4 (1–7)	1 (0–1)	0.006
Per 100,000 patient admissions	0.21	0.03	0.010
Per 10,000 blood culture requests	0.14	0.02	0.006
Methicillin-sensitive *Staphylococcus aureus*			
Total number of blood cultures	2963	833	
Mean (range) blood cultures per year	370 (308–464)	417 (397–436)	0.081
Per 100,000 patient admissions	21.73	24.46	0.001
Per 10,000 blood culture requests	14.59	15.45	0.042
Methicillin-resistant *Staphylococcus aureus*			
Total number of blood cultures	3498	1012	
Mean (range) blood cultures per year	437 (353–538)	506 (455–557)	0.094
Per 100,000 patient admissions	25.66	29.72	<0.001
Per 10,000 blood culture requests	17.22	18.78	0.179
*Escherichia coli*			
Total number of blood cultures	8034	1075	
Mean (range) blood culture per year	1004 (816–1125)	988 (963–1012)	0.695
Per 100,000 patient admissions	58.93	58.00	0.807
Per 10,000 blood culture requests	39.55	36.64	0.021

^a^ Blood culture collected > 2 days after hospital admission and there is no case of *Neisseria meningitidis* fulfilling the definition of hospital-onset bacteremia.

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
