# Peer review of "Decreased Antibiotic Consumption Coincided with Reduction in Bacteremia Caused by Bacterial Species with Respiratory Transmission Potential during the COVID-19 Pandemic"

_antibiotics, 2022, doi:10.3390/antibiotics11060746_

Round 1

Reviewer 1 Report

The authors have done a great job in compiling extensive data to answer their study question. The study is well performed and written. The inferences drawn from results and their explanation seem acceptable.

Author Response

Reply to reviewer 1

Comments and Suggestions for Authors

The authors have done a great job in compiling extensive data to answer their study question. The study is well performed and written. The inferences drawn from results and their explanation seem acceptable.

Ans: Thank you for your positive comment.

Reviewer 2 Report

“Reduction in Antibiotic Consumption Coincided with Significant Reduction in Bacteremia due to Bacterial Species with Potential of Respiratory Transmission when Non-Pharmaceutical Intervention against Respiratory Infections was Instituted during the COVID-19 Pandemic”

By Cheng et al.

In this paper, the authors show that non-pharmaceutical interventions such as masking and hand hygiene during the Covid-19 pandemic, significantly decreased total antibiotic consumption and incidence of bacteremia caused by airborne infections.  The conclusion is based on analysis of a very large dataset from 43 public hospitals that together provide 90% of in-patient services in Hong Kong. While this result is not surprising and can be expected, this report provides evidence for such a correlation.  This reduction of antibiotic consumption took place in spite of the fact that many Covid-19 patients were treated with antibiotics with no clinical evidence of bacterial co-infection.

The manuscript has been mostly well written and is an important contribution in the field.  I have the following minor comments on the manuscript:

Title, Line 2: I had to read the title several times to understand what it means. I think the title should be reworded.  The reader will be confused because of two apparent statements in the title: (1) That the coincidence of reduction in antibiotic consumption and reduction in bacteremia is the big discovery in this paper, which is actually not what the authors mean. (2) “Reduction in bacteremia due to bacterial…” Does the “due to” refer to the reduction or the bacteremia? The confusion can be cleared by using a different language such as “Reduction in bacteremia caused by bacterial…”. The phrase, “bacteremia due to” has been used in the manuscript 15 times.  It is only in the title that this confusion arises because it is preceded by “reduction in”.

Line 35 and Line 84: Change “wholesales supply” to “wholesale supply”

Line 106: Change “indicator on” to “indicator of”

All the figures (1,2,3,4,5): Both the X-axis, Y-axis, the scale marks, and the descriptions of the different bars and lines are all very faint and in small fonts. They are very difficult to read especially when black-and-white copies are printed.  Please make the lines thicker and the fonts larger.

Lines 171-172: The different items mentioned are not in the same order as in Fig 2.  I understand that the authors have written them in decreasing order of percentages, but I think, they should match the order in the figure for easier reading.  Or, the authors can change the order in the figure.

Figure 2: While all other figures and tables include data from 2021, I wonder why figure 2 does not include 2021.

Figure 4: The graph is not serving its purpose. Some of the bars (orange and dark blue) are too small to understand the trend. The authors can think of a better way of displaying the results. One suggestion is to draw line graphs for all, as is already done for E. coli. There can be three panels (one above the other) with the same X-axis. This way the Y-axes can have different scales: one from 0 to 200, one from 400 to 800 and one from 4500 to 6500.

Line 264: “over17” Add space between over and 17

Line 293: “colonization rate” If possible, define the term in a few words.  Since you are providing percentages, the reader should know what these numbers mean.

Line 297: “incidence of…bacteremia…improved…” The word “improved” is confusing.  Improvement for the bacteria or for the host? It will be better to say “increased” or “decreased”

Line 306: “influenza usually start”.  I guess, influenza is singular. Then it should be “influenza usually starts”

Line 371: “per 100,000 patient admissions per year” and “per 10,000 blood culture requests per year” have been written in the manuscript numerous times. These are units of measurement and counting.  The unit can be mentioned when numbers are presented or the method is described but not every time “bacteremia” is mentioned.

Line 382: A “Conclusion” section will probably be appropriate.  There is so much data presented in the manuscript, a conclusion section with the important points will help the reader understand the significance of the paper.

Author Response

Reply to reviewer 2

Comments and Suggestions for Authors

“Reduction in Antibiotic Consumption Coincided with Significant Reduction in Bacteremia due to Bacterial Species with Potential of Respiratory Transmission when Non-Pharmaceutical Intervention against Respiratory Infections was Instituted during the COVID-19 Pandemic”

By Cheng et al.

In this paper, the authors show that non-pharmaceutical interventions such as masking and hand hygiene during the Covid-19 pandemic, significantly decreased total antibiotic consumption and incidence of bacteremia caused by airborne infections.  The conclusion is based on analysis of a very large dataset from 43 public hospitals that together provide 90% of in-patient services in Hong Kong. While this result is not surprising and can be expected, this report provides evidence for such a correlation.  This reduction of antibiotic consumption took place in spite of the fact that many Covid-19 patients were treated with antibiotics with no clinical evidence of bacterial co-infection.

The manuscript has been mostly well written and is an important contribution in the field.  I have the following minor comments on the manuscript:

Title, Line 2: I had to read the title several times to understand what it means. I think the title should be reworded.  The reader will be confused because of two apparent statements in the title: (1) That the coincidence of reduction in antibiotic consumption and reduction in bacteremia is the big discovery in this paper, which is actually not what the authors mean. (2) “Reduction in bacteremia due to bacterial…” Does the “due to” refer to the reduction or the bacteremia? The confusion can be cleared by using a different language such as “Reduction in bacteremia caused by bacterial…”. The phrase, “bacteremia due to” has been used in the manuscript 15 times.  It is only in the title that this confusion arises because it is preceded by “reduction in”.

Ans: Thank you for your suggestion.

In our submitted version, the manuscript title is “Reduction in antibiotic consumption coincided with significant reduction in bacteremia due to bacterial species with potential of respiratory transmission when non-pharmaceutical intervention against respiratory infections was instituted during the COVID-19 pandemic”

We have revised the manuscript title as follow:

“Decreased antibiotic consumption coincided with reduction in bacteremia caused by bacterial species with respiratory transmission potential during the COVID-19 pandemic”

Line 35 and Line 84: Change “wholesales supply” to “wholesale supply”

Ans: Thank you for your suggestion. The change is made accordingly.

Line 106: Change “indicator on” to “indicator of”

Ans: Thank you for your suggestion. The change is made accordingly.

All the figures (1,2,3,4,5): Both the X-axis, Y-axis, the scale marks, and the descriptions of the different bars and lines are all very faint and in small fonts. They are very difficult to read especially when black-and-white copies are printed.  Please make the lines thicker and the fonts larger.

Ans: Thank you for your suggestion. We have increased the font size from 9 to 14. The width of line is increased from 0.75 to 2 in all figures.

Lines 171-172: The different items mentioned are not in the same order as in Fig 2.  I understand that the authors have written them in decreasing order of percentages, but I think, they should match the order in the figure for easier reading.  Or, the authors can change the order in the figure.

Ans: It is correct that we would like to present the data in decreasing order in our submitted version.

“During the COVID-19 pandemic, a reduction of antibiotic supply to private doctors (37.7%), community pharmacies (28.8%), private hospitals (29.0%), and Hospital Authority (10.3%) was observed between 2019 and 2020 in Hong Kong.”

We follow your suggestion to present the data according to the order in the figure for easier reading. In addition, we have included data from 2021.

“During the COVID-19 pandemic, a reduction of antibiotic supply to private doctors (39.0%), community pharmacies (47.9%), private hospitals (35.1%), and Hospital Authority (12.5%) was observed between 2019 and 2021 in Hong Kong.”

Figure 2: While all other figures and tables include data from 2021, I wonder why figure 2 does not include 2021.

Ans: Thank you for your suggestion. It is because the data from 2021 is not yet available from the HKSAR government at the time of manuscript preparation. However, we just obtain the data from 2021 and include in the revised manuscript.  

Figure 4: The graph is not serving its purpose. Some of the bars (orange and dark blue) are too small to understand the trend. The authors can think of a better way of displaying the results. One suggestion is to draw line graphs for all, as is already done for E. coli. There can be three panels (one above the other) with the same X-axis. This way the Y-axes can have different scales: one from 0 to 200, one from 400 to 800 and one from 4500 to 6500.

Ans: Thank you for your suggestion. We have revised Figure 4 according to the suggestion of using three panels with the same X-axis and Y-axes with different scales.

Line 264: “over17” Add space between over and 17

Ans: Thank you for your suggestion. The change is made accordingly.

Line 293: “colonization rate” If possible, define the term in a few words.  Since you are providing percentages, the reader should know what these numbers mean.

Ans: Thank you for your suggestion. The change is made accordingly.

In our submitted version, it is stated that “Since the nasal colonization rate of S. aureus and gut colonization rate of E. coli are around 30% and 100% in the adult population respectively”

To allow better understanding of the reader, we have rephrased the sentence by removing the technical term of “colonization rate”. The revised sentence is listed below:

“Since the nasal carriage of S. aureus and gut carriage of E. coli are around 30% and 100% in the adult population respectively”

Line 297: “incidence of…bacteremia…improved…” The word “improved” is confusing.  Improvement for the bacteria or for the host? It will be better to say “increased” or “decreased”

Ans: Thank you for your suggestion. The change is made accordingly.

In our submitted version, it is stated that “This may partially explain the findings that the incidence of S. aureus and E. coli bacteremia were not significantly improved….”

We have changed the term “improved” to “decreased”.

“This may partially explain the findings that the incidence of S. aureus and E. coli bacteremia were not significantly decreased….”

Line 306: “influenza usually start”.  I guess, influenza is singular. Then it should be “influenza usually starts”

Ans: Thank you for your suggestion. The change is made accordingly.

Line 371: “per 100,000 patient admissions per year” and “per 10,000 blood culture requests per year” have been written in the manuscript numerous times. These are units of measurement and counting.  The unit can be mentioned when numbers are presented or the method is described but not every time “bacteremia” is mentioned.

Ans: Thank you for your suggestion. The change is made accordingly.

In our submitted version, it is stated that “There are several limitations of this study. Firstly, we do not completely understand the finding of a significant increase in community-onset MRSA bacteremia per 100,000 patient admissions per year during the COVID-19 pandemic,”

We have removed the “per 100,000 patient admissions per year” in the sentence.

“There are several limitations of this study. Firstly, we do not completely understand the finding of a significant increase in community-onset MRSA bacteremia during the COVID-19 pandemic,” 

Line 382: A “Conclusion” section will probably be appropriate.  There is so much data presented in the manuscript, a conclusion section with the important points will help the reader understand the significance of the paper.

Ans: Thank you for your suggestion. We have added a Conclusion section in the revised manuscript.

In summary, the non-pharmacological interventions during the last two pandemic years of COVID-19 in Hong Kong has markedly reduced the incidence of acute respiratory virus and bacterial infections transmitted by the droplet and airborne route. This change is most evident in the incidence of community-onset bacteremia due to S. pyogenes, S. pneumoniae, H. influenzae and N. meningitidis, and is independently confirmed by the marked decrease in the notification of scarlet fever, chickenpox, and tuberculosis, which are largely droplet or airborne infections. Notably this change has coincided with the decreased in our wholesale supply of antibiotics. Our findings suggested that non-pharmaceutical measures in reducing the incidence of respiratory infections may be one of the strategies to control the amount of antibiotic consumption in human which may impact on the burden of antibacterial resistance in the long run.

Reviewer 3 Report

The title is too long and needs to be shortened into a simpler one. The observation of the reduction in wholesale supply of antibiotics in the Abstract is relevant however it should be moved in the Introduction section. The reduction of infection with airborne pathogens should be rather linked with mandatory mask usage. The abstract should be re-written, stating the brief introduction and scope of the research, materials and methods results and conclusions.

Line 66: "our" should be taken out. simply state that hospital infection control measures were enforced.

Data regarding how Streptococcus pneumoniae[1], Haemophilus influenzae [2] and Neisseria meningitidis commonly spread should be added

Data regarding common vectors for S. aureus and E. coli must be added [4]

Statistical analysis is lacking, please find solid correlation points and perform the analysis

References

  1. Weiser, J. N., Ferreira, D. M., & Paton, J. C. (2018). Streptococcus pneumoniae: transmission, colonization and invasion. Nature reviews. Microbiology16(6), 355–367. https://doi.org/10.1038/s41579-018-0001-8
  2. Lee MH, Lee GA, Lee SH, Park YH (2020) A systematic review on the causes of the transmission and control measures of outbreaks in long-term care facilities: Back to basics of infection control. PLOS ONE 15(3): e0229911. https://doi.org/10.1371/journal.pone.0229911
  3.  Cozorici, D., Măciucă, R. A., Stancu, C., Tihăuan, B. M., Uță, R. B., Codrea, C. I., Matache, R., Pop, C. E., Wolff, R., & Fendrihan, S. (2022). Microbial Contamination and Survival Rate on Different Types of Banknotes. International journal of environmental research and public health19(7), 4310. https://doi.org/10.3390/ijerph19074310

Author Response

Reply to Reviewer 3

Comments and Suggestions for Authors

The title is too long and needs to be shortened into a simpler one.

Ans: Thank you for your suggestion. We have revised the manuscript title as follow:

“Decreased antibiotic consumption coincided with reduction in bacteremia caused by bacterial species with respiratory transmission potential during the COVID-19 pandemic”

The observation of the reduction in wholesale supply of antibiotics in the Abstract is relevant however it should be moved in the Introduction section.

Ans: Thank you for your suggestion.

We have included the observation of the reduction in wholesale supply of antibiotics in the Introduction section.

“we observed a reduction in wholesale supply of antibiotics in both the community and hospital settings which coincided with a significant reduction in bacteremia caused by encapsulated bacteria with the potential of respiratory transmission”

The reduction of infection with airborne pathogens should be rather linked with mandatory mask usage. The abstract should be re-written, stating the brief introduction and scope of the research, materials and methods results and conclusions.

Ans: Thank you for your suggestion.

We have revised the abstract and clearly stated the introduction and scope of the research, materials and methods, results and conclusions.

Line 66: "our" should be taken out. simply state that hospital infection control measures were enforced.

Ans: Thank you for your suggestion. The change has been made accordingly.

Data regarding how Streptococcus pneumoniae[1], Haemophilus influenzae [2] and Neisseria meningitidis commonly spread should be added. Data regarding common vectors for S. aureus and E. coli must be added [4]

References

  1. Weiser, J. N., Ferreira, D. M., & Paton, J. C. (2018). Streptococcus pneumoniae: transmission, colonization and invasion. Nature reviews. Microbiology16(6), 355–367. https://doi.org/10.1038/s41579-018-0001-8
  2. Lee MH, Lee GA, Lee SH, Park YH (2020) A systematic review on the causes of the transmission and control measures of outbreaks in long-term care facilities: Back to basics of infection control. PLOS ONE 15(3): e0229911. https://doi.org/10.1371/journal.pone.0229911
  3.  Cozorici, D., Măciucă, R. A., Stancu, C., Tihăuan, B. M., Uță, R. B., Codrea, C. I., Matache, R., Pop, C. E., Wolff, R., & Fendrihan, S. (2022). Microbial Contamination and Survival Rate on Different Types of Banknotes. International journal of environmental research and public health19(7), 4310. https://doi.org/10.3390/ijerph19074310

Ans: Thank you for your suggestion as well as the suggested references.

In our discussion, we have mentioned the common route of spread for Streptococcus pneumoniae, Haemophilus influenzae and Neisseria meningitidis, as well as for S. aureus and E. coli.

We have included 3 additional references as suggested:

S. aureus and E. coli are well known to be transmitted through direct and indirect contact of patients, food and environment, including contaminated banknotes [ref: Cozorici D, Măciucă RA, Stancu C, Tihăuan BM, Uță RB, Codrea CI, et al. Microbial Contamination and Survival Rate on Different Types of Banknotes. Int J Environ Res Public Health. 2022 Apr 4;19(7):4310.], whereas S. pyogenes, S. pneumoniae, H. influenzae, and N. meningitidis are long believed to be transmitted by droplets or direct/indirect contact with respiratory secretion [Weiser JN, Ferreira DM, Paton JC. Streptococcus pneumoniae: transmission, colonization and invasion. Nat Rev Microbiol. 2018 Jun;16(6):355-367.; Lee MH, Lee GA, Lee SH, Park YH. A systematic review on the causes of the transmission and control measures of outbreaks in long-term care facilities: Back to basics of infection control. PLoS One. 2020 Mar 10;15(3):e0229911.].”

Statistical analysis is lacking, please find solid correlation points and perform the analysis

Ans: Thank you for your suggestion.

We have added a sub-section of statistical analysis in the Method section. We also consulted a statistician, Dr. Pui-Hing Chau for the use of statistical method. We have changed our statistical method from Student t test to Poisson Regression in the revised version.

“Statistical analysis

“Differences in the magnitude of wholesale supply of antibiotics expressed as defined daily doses per 1,000 inhabitants per day, as well as community-onset and hospital-onset bacteremia due to S. pyogenes, S. pneumoniae, H. influenzae, MSSA, MRSA, and E. coli in terms of number and rates per 100,000 patient admissions, and per 100,000 blood culture requests, were evaluated between period 1 (2012 to 2019) and period 2 (2020 to 2021) using Poisson Regression. All statistical analyses were performed using IBM SPSS Statistics (version 26). A two-sided p-value of < 0.05 was considered statistically significant.” 

Reviewer 4 Report

It is an interesting study that allows us to have an overview of the consumption of antibiotics and bacteremia mainly respiratory infections.
The charts cover a large time range that allow us to discern the fluctuations.
However i would suggest to authors to change the title to a smaller one.
In conclusion, I believe that it can be published as it is as it is interesting information about the COVID pandemic and the use of antibiotics.

Author Response

Reply to reviewer 4

Comments and Suggestions for Authors

It is an interesting study that allows us to have an overview of the consumption of antibiotics and bacteremia mainly respiratory infections.
The charts cover a large time range that allow us to discern the fluctuations.
However i would suggest to authors to change the title to a smaller one.
In conclusion, I believe that it can be published as it is as it is interesting information about the COVID pandemic and the use of antibiotics.

Ans: Thank you for your suggestion. We have revised the manuscript title as below:

“Decreased antibiotic consumption coincided with reduction in bacteremia caused by bacterial species with respiratory transmission potential during the COVID-19 pandemic”

Round 2

Reviewer 3 Report

The article has been extensively revised and I find it suitable for publication after a minor revision.

Abstract should not contain the words "Introduction; Materials and Methods:; Results; Conclusions" those are simply chapter titles.

Statistical analysis is good.

The theory that "non-pharmaceutical measures in reducing the incidence of respiratory infections may be one of the strategies to control the amount of antibiotic consumption in human which may impact on the burden of antibacterial resistance in the long run" is supported through the manuscript